# FEM-Based Power Transformer Model for Superconducting and Conventional Power Transformer Optimization

Tamás Orosz 

Department of Power Electronics and Electrical Drives, Széchenyi István University, Egyetem tér 1, 9026 Győr, Hungary; orosz.tamas@sze.hu

**Abstract:** There were many promising superconducting materials discovered in the last decades that can significantly increase the efficiency of large power transformers. However, these large machines are generally custom-made and tailored to the given application. During the design process the most economical design should be selected from thousands of applicable solutions in a short design period. Due to the nonlinearity of the task, the cost-optimal transformer design, which has the smallest costs during the transformers' planned lifetime, is usually not the design with the highest efficiency. Due to the topic's importance, many simplified transformer models were published in the literature to resolve this problem. However, only a few papers considered this preliminary design optimization problem in the case of superconducting transformers and none of them made a comparison with a validated conventional transformer optimization model. This paper proposes a novel FEM-based two-winding transformer model, which can be used to calculate the main dimension of conventional and superconducting transformer designs. The models are stored in a unified JSON-file format, which can be easily integrated into an evolutionary or genetic algorithm-based optimization. The paper shows the used methods and their accuracy on conventional 10 MVA and superconducting 1.2 MVA transformer designs. Moreover, a simple cost optimization with the 10 MVA transformer was performed for two realistic economic scenarios. The results show that in some cases the cheaper, but less efficient, transformer can be the more economic.

**Keywords:** power transformers; optimization; finite element analysis; evolutionary algorithms; superconductors

## 1. Introduction

The European Commission introduced the Ecodesign directive in 2009 [1]. This regulation aims to decrease the losses and the $CO_2$ emission in the electrical grid by limiting the lifetime losses for all kinds of transformers [1,2]). The High-Temperature Superconductor (HTS)-based transformers can help to achieve this goal in the future.

Finding the optimal losses and minimizing the lifetime costs is a fundamental question in the transformer literature from the beginning of the industry. The first papers, analytical calculations, were published more than a hundred years ago on this topic [3]. Nowadays, large power transformers are highly optimized designs, tailored to the specific application [3–5]. The transformers' lifetime cost is usually calculated by the capitalized costs of the losses and the production price of the transformer. For a quotation, usually, a preliminary or conceptional transformer design is used, where the transformer is modelled by only its key design parameters (Figure 1) [4,6]. In the case of electrical grid transformers, the different production curves and prices of the different power plants result from different capitalization costs, which result from different cost-optimal design parameters. Therefore, the exact technical requirements and the cost-optimal transformer design should be selected from thousands of good designs [3,6,7].

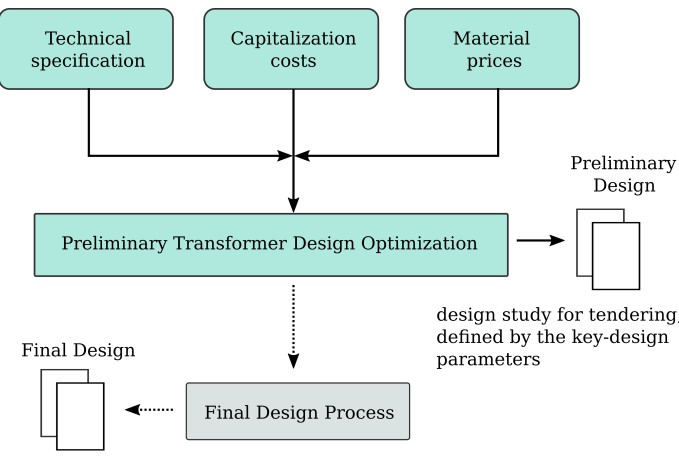

**Figure 1.** The place of the preliminary or conceptional design and its inputs during a custom manufacturing process.

Application of modern HTS materials in power transformers can have many advantages, such as lower load losses, smaller size and short-circuit current reducing capability [8–13]. Only a few years after the development of the first Rare-Earth (Re) Barium Copper Oxide (ReBCO, where Re can represent Yttrium (Y), Gadolinium (Gd), Samarium (Sm) or a combination of these materials) high-temperature superconducting cables, many superconducting transformer projects have been initiated (Table 1). In 1997, ABB built the first prototype transformer in Switzerland, which was connected to the power grid and operated for a year successfully [10,14–16]. These projects successfully applied liquid nitrogen-cooled ReBCO cables in distribution transformer prototypes up to 6.5 MVA (Table 1). Many HTS transformers were successfully tested, and some operated for more than a year in real working conditions [11,17].

**Table 1.** Non-exclusive list of superconducting transformer projects from 1997 until today [11].

| Year | Country | Transformer |
|------|---------|-------------|
| 1997 | Switzerland | 630 kVA, three-phase, $U_{hv}/U_{lv}$ = 18.72 kV/0.42 kV |
| 1997 | Japan | 5 kVA, three-phase, $U_{hv}/U_{lv}$ = 6.6 kV/3.3 kV |
| 1998 | USA | 1 MVA, single-phase, $U_{hv}/U_{lv}$ = 13.8 kV/6.9 kV |
| 1999 | Korea | 100kVA, three-phase, $U_{hv}/U_{lv}$ = 440 V/220 V |
| 2001 | Germany | 1 MVA, single-phase, $U_{hv}/U_{lv}$ = 25 kV/1.4 kV |
| 2003 | Germany | 100 kV, single-phase, $U_{hv}/U_{lv}$ = 5.5 kV/1.1 kV |
| 2004 | Japan | 4 MVA, single-phase, $U_{hv}/U_{lv}$ = 25 kV/1.2 kV |
| 2005 | France | 41 kVA, single-phase, $U_{hv}/U_{lv}$ = 2.05 kV/0.41 kV |
| 2010 | New Zeeland | 1 MVA, three-phase, $U_{hv}/U_{lv}$ = 11 kV/0.4 kV |
| 2010 | Japan | 2 MVA, three-phase, $U_{hv}/U_{lv}$ = 66 kVA/6.9 kV |
| 2010 | Italy | 10kVA, single-phase, $U_{hv}/U_{lv}$ = 1 kV/230 V |
| 2013 | Japan | 2 MVA, three-phase, $U_{hv}/U_{lv}$ = 66 kV/6.9 kV |
| 2014 | Japan | 400 kVA, single-phase, $U_{hv}/U_{lv}$ = 6.9 kV/2.3 kV |
| 2016 | Russia | 330 kVA, single-phase, $U_{hv}/U_{lv}$ = 10 kV/231 V |
| 2017 | China | 330 kVA, single-phase, $U_{hv}/U_{lv}$ = 10 kV/231 V |
| 2019 | Switzerland | 577 kVA, single-phase, $U_{hv}/U_{lv}$ = 20 kV/1 kV |
| 2020 | China | 6.5 MVA, single-phase, $U_{hv}/U_{lv}$ = 25 kV/1.9 kV |

Scaling up and comparing the economic competitiveness of large HTS transformers with conventional transformers needs the accurate modelling of AC losses in ReBCO tapes. Many analytical and numerical methods worked out and tested in the last decades for accurate modelling of AC losses in superconducting windings [18–25]. These methods usually categorize the major source of HTS winding losses into the following categories:

hysteresis losses, eddy current losses and coupling losses [26]. Therefore, the winding losses are significantly affected by the magnetic field distribution.

Baldwin et al. [27] published one of the first preliminary design models for a 1.167 MVA, superconducting, shipboard transformer [12]. One of the main conclusions of this paper is that the transformer losses can be significantly reduced by applying a proper winding layout, which can reduce the radial component of the magnetic induction at the winding ends. The authors used analytical formulas [28,29] to approximate the maximal value of the axial and radial components of the magnetic induction in the windings. Morandi et al. [30] also used this analytical approximation for design optimization and cost predictions for a 25 MVA HTS transformer. The loss predictions in the above-mentioned two models used only the maximal value of the radial and axial fluxes in the loss prediction [31]. Therefore, it can significantly overestimate the transformer losses because the axial flux has a maximum at the center of the winding and a minimum at the winding ends, where the radial flux has its maximum.

The authors did not define a minimal value for the short-circuit impedance in these papers. The resulting designs have lower short circuit impedance than the conventional transformer standards. This kind of comparison of the conventional and HTS designs is not correct because decreasing the short-circuit impedance can significantly reduce the total cost of ownership in the case of conventional transformers as well [32]. These relatively small leakage inductances result in higher short circuit forces, where the brittleness of the ReBCO materials and the short circuit limiting the capacity of the superconductor cable should be considered [33,34]. Berger et al. [35] compared the efficiency and the losses of a 63 MVA HTS transformer. They have found that the superconducting transformer has raised efficiency. The problem with this kind of comparison is that the two transformers are not selected on the same basis. The optimal efficiency with the given technology depends on the applied capitalization cost of the transformer [7], but this effect was not considered in their paper.

Staines et al. created a more sophisticated comparison on 40 MVA conventional and superconducting transformers [36]. The authors used 50 \$/kAm wire prices for the analysis, and they have found that the Cryocooler and the HTS wire prices should be decreased to be interesting for a typical substation transformer application. This estimation is more optimistic than some older prognostics, which calculated 10 \$/kA/m for market entry price criterion [37–39]. Moreover, the lower weight and the higher power/mass ratio can make these transformers interesting for applications with strict space and mass limitations.

The paper proposes a FEM-based two-winding transformer model, which can be applied to calculate the main design parameters of a conventional power transformer and an HTS transformer. This transformer model is the extension of the widely used model, where the different winding layouts are modelled by their copper filling factor [6,40]. The transformer models can be described as key-value pairs in a JSON file, while the optimizer can automatically build up and perform the required analytical and FEM-based calculations. The TrafoCalc library uses the integrated, hp-adaptive FEM solver Agros Suite to automatically build a transformer model and determine the magnetic field during the working conditions [41]. It can estimate the short circuit impedance, the tensile stresses, and the axial and radial flux distribution along the windings.

The similarity of the models can ensure that the transformers are compared on the same basis [40]. Moreover, the proposed calculations were validated on the published data of manufactured transformers, and they were integrated into the Python-based Trafo-Calc library. The source code of the project and proposed calculations published in the GitHub repository of the project (https://github.com/tamasorosz/TrafoCalc, accessed on 28 May 2022). The proposed calculations can be openly used as simple benchmarks for other transformer design problems.

## 2. Materials and Methods

### 2.1. FEM-Based Two-Winding Transformer Model

Many simplified transformer models have been published to resolve the preliminary design optimization problem. Many of them model the transformer by its core and the windings (active part) because its size approximates well the total cost of the machine [3,4,29]. This paper uses a similar two winding active part model to approximate the dimensions of both the conventional and the superconducting transformer. However, in the case of superconducting transformers, the external cooling system has a significant impact on the price of the product and power consumption. The proposed algorithm of the TrafoCalc library can consider this, and it uses the hp-adaptive FEM solver Agros Suite to simulate the magnetic field density in the transformer's working window. In a conventional transformer, the load losses, short circuit forces, and the short circuit impedance are estimated from the magnetic field distribution of the working window, both in the case of conventional and superconducting transformers.

The implemented algorithm can handle only three-phased core-form power transformers with a three-legged core. It can consider two-winding designs or two-winding designs where the regulation winding is built-in one of the transformer windings. Every transformer design can be defined by its required and independent parameters. These parameters are stored in JSON file format as key-value pairs in a separate directory, and the proposed code can automatically perform the detailed calculations for every distinct transformer design. Every transformer design can be defined by six parameters (Table 2). These parameters can be handled as genes of an individual, which any evolutionary or genetic algorithm can optimize [40,42]. The three dimensional view of the modeled transformer design is plotted in Figure 2, while the meaning of these optimized parameters is discussed in Table 2. The following sections show the detailed calculation of the losses, short-circuit impedance and other required parameters of the designed transformer.

**Table 2.** The independent parameters of the applied two-winding transformer model in the case of conventional and superconducting transformers.

| Parameter | Dimension | Notation |
|---|---|---|
| Core diameter | mm | $D_c$ |
| Magnetic flux density | T | $B_c$ |
| Height of the low voltage winding | mm | $h_{lv}$ |
| Current density in the low voltage winding | $\frac{A}{mm^2}$ | $j_{lv}$ |
| Current density in the high voltage winding | $\frac{A}{mm^2}$ | $j_{hv}$ |
| Main insulation distance | mm | $g$ |

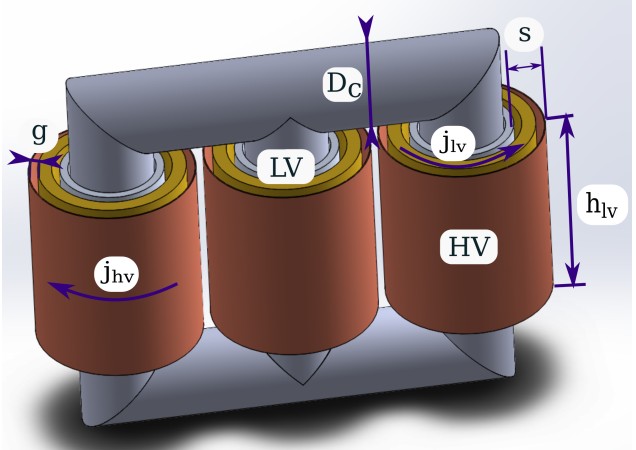

**Figure 2.** The optimized three-phase transformer model and its independent parameters.

## 2.2. Objective Function

The total ownership cost (*TCO*) is used as the objective of the optimization. The calculation of this measure is well-defined for power transformers in the manufacturing standards [43] and widely studied by many researchers [3,40,44,45]. This quantity is calculated by the sum of the estimated price of the product and the cost of the lifetime losses. This function is the same for the conventional and the superconducting transformer, the remarkable cost of the superconducting transformer's cooling cost is considered by additional load loss cost [46]:

$$TCO = k_1 \cdot P_{nll} + k_2 \cdot P_{ll} + P_p. \tag{1}$$

where $k_1$ and $k_2$ denote the capitalized cost of the no-load the load losses in EUR/kW for its planned lifetime, which is usually 18–30 years. These costs consider the fix and the variable costs of the commission, losses, maintenance and installation during the transformers lifecycle [43]. The capitalization cost values are multiplied by the transformer's load and the no-load losses, denoted by $P_{ll}$ and $P_{nll}$ and calculated in kW. Product price $P_p$ is calculated differently for the conventional and superconducting transformers in the applied model.

## 2.3. Manufacturing Cost

In case of a conventional transformer the product price is usually approximated by the size of the core and the windings, whose size approximates well the total size of the machine, which contains the bushings, cooling equipment, and the different accessories:

$$P_p = \lambda \cdot \sum_{i=0}^{n} c_i \cdot M_i, \tag{2}$$

where $\lambda$ represents the approximate ratio of the transformer active part and the total cost of the manufactured transformer. $M_i$ is the mass of the $i$th part of the active part model ($i$ index marks the core ($i = 0$), LV ($i = 1$) and HV ($i = 2$) winding) in kg and $C_i$ represents the specific cost of the denoted part of the transformer in EUR/kg. The $\lambda$ factor can estimate the cost of the bushings, radiators and other accessories, which cost is well determined by the requirements and the active part size.

In the case of high-temperature superconducting transformers, the cooling cost has a crucial role in the final cost and the competitiveness of the final design. Their cost ($P_{cp}$) can be calculated by the following formula [46]:

$$P_{cp} = 1.81 \cdot P_c^{0.57} \cdot 1000, \tag{3}$$

where the $P_c$ approximates the required cooling power in kW, this cooling power should be higher than the total load loss of the windings, current leads and the cryostat. This formula should be updated for the current prices. However, this price was never considered during the analysis and comparison of superconducting and conventional transformers.

## 2.4. Core Mass and No-Load Losses

Due to the iron core not being cooled in a superconducting transformer, the core losses and the applied technology are considered the same as conventional core-form transformers. The dimensions of the cryostat cause the only difference between the two cases. Due to the size of the cryostat, a higher insulation level should be applied between the core and the inner windings, as for conventional transformers. This difference can imply that, in some cases, the insulation distances can be minimized by applying an unusual winding order—the high voltage winding placed in the inner position—as in the case of large autotransformers [6].

The modeled transformer core can be calculated by the sum of the following parts of the core [3,6,47] :

$$M_c = M_{leg} + M_{yoke} + M_{corner}, \tag{4}$$

where the $M_c$ represents the total mass of the core in kg units, $M_{leg}$ means the total mass of the three legs, $M_{yoke}$ represents the total mass of the yokes and $M_{corner}$ represents the mass of the transformer corners. In these corners, the sheets are overlapping, which is considered by simple building factor due to the different manufacturing technology. The following way calculates these parts of the modelled, three-legged core:

$$M_{corner} = 8 \cdot R_c^3 \cdot \lambda_c \cdot \pi \cdot \rho_{fe}, \tag{5}$$

$$M_{leg} = R_c^2 \cdot \lambda_c \cdot \pi \cdot \rho_{fe} \cdot (EI + h_{lv}), \tag{6}$$

$$M_{yoke} = R_c^2 \cdot \lambda_c \cdot \pi \cdot \rho_{fe} \cdot (4 \cdot s + 2 \cdot p_d + 6 \cdot R_c), \tag{7}$$

where $\lambda_c$ means the filling factor of the core, the calculation of this quantity is based on the lamination of the applied electrical steel and manufacturing technology. The $\rho_{fe}$ represents the density of the electrical steel. The sum of the low voltage winding ($h_{lv}$) height and the end insulation ($EI$) distance is used to calculate the height of the core window. In the case of the superconducting transformers, the $EI$ distance contains the approximate thickness of the cryostat as well. The width of the transformer's working window is calculated from the sum of the window width $s$ and the value of the phase distance $p_d$ [6,40].

The core loss ($P_{nll}$) is calculated by the fitted unit price of the applied electrical steel (Figure 3) [6,29,47]:

$$P_{nll} = f_b \cdot p_{nll} \cdot M_c, \tag{8}$$

where $p_{nll}$ is the unit loss of the applied electrical steel (W/kg) (Figure 3) and $f_b$ is the building factor, which depends on the applied manufacturing technology. In our calculations, it is considered to be 1.2 [48].

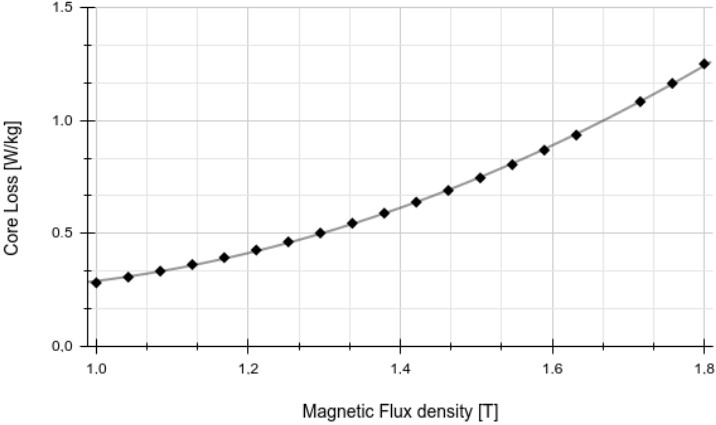

**Figure 3.** The built-in unit loss function of a 27HP110 grade steel for the calculations at 50 Hz. The lamination thickness of the applied steel sheet is 0.27 mm and the core loss at 1.7 is 1.05 W/kg [49].

*2.5. Load Losses*

Conventional Transformers

The load loss calculation in the conventional transformer model simply calculated from the sum of the DC and the AC losses. The DC loss is calculated simply from the mass of the windings using its copper filling factor based dimensions:

$$P_{loss} = P_{dc} + \kappa P_{ac}, \tag{9}$$

$$P_{dc} = \rho \cdot \frac{2 \cdot \pi \cdot r_m}{A_{cu}} \cdot I^2, \tag{10}$$

where $\rho$ represents the conductor's specific conductivity, $r_m$ denotes the mean radius, $A_{cu}$ is the total area of the copper in the winding, and $I$ is the phase current of the examined

coil in Amps. The $\kappa$ is an empirical factor, generally used to consider the tank's generated losses and the transformer's other structural part.

The eddy losses, which are generated in the windings, are approximated by simple empirical constants, which are given by the outer dimensions, the applied filling factor, and the type of the winding:

$$P_{eddy} = \kappa_2 P_{dc}, \tag{11}$$

where $\kappa_2$ represents the value of the eddy-loss factor. More precise calculations can be used by using the FEM-based magnetic field calculations, where the eddy losses can be calculated using the radial ($B_{rad}$) and axial ($B_{ax}$) components of the magnetic induction [40,42]:

$$P_{ax} = \frac{1}{24\rho}(B_{ax} \cdot \omega \cdot d^*)^2, \tag{12}$$

$$P_{rad} = \frac{1}{24\rho}(B_{rad} \cdot \omega \cdot h^*)^2, \tag{13}$$

in these formulas, $w$ represents the width of the elementary turn in the designed winding layout, while $h$ is the turn's height in mm, $\omega = \pi f$ is the circular frequency, while $\rho$ is the electrical conductivity of the applied conductor. The eddy current losses of a winding can be calculated by the sum of the axial and the radial losses (Equations (12) and (13)). This calculation uses the copper filling factor-based winding model to estimate the magnetic flux density in the winding segments. Different segments can model the windings with different filling factors during the FEM modelling. This calculation assumes that the conductors do not modify the magnetic field of the winding segments. Using the above-mentioned simplifications, this methodology can estimate the losses in the windings with a good agreement of the measured values in the case of different winding types [29,48].

### 2.6. Load Loss Calculation in Superconducting Transformers

The load loss estimation is significantly differs in the superconducting case. The DC losses in the superconducting windings can be neglected, while the thermal loss of the cryostat and the current leads should be considered together with the AC losses of the windings [30]:

$$P_{ll} = C_{cf} \cdot (P_{ac} + P_{cr} + P_{cl}), \tag{14}$$

where $C_{cf}$ is an empirical factor for considering the losses of the applied cooling system, due to the applied references [30], the worst efficiency of the coolers is around 5% while the best cooler efficiency is around 15%, so the $C$ value should be selected between 20 and 12 [46]. $P_{ac}$ represents the AC losses of the transformer windings, $P_{cr}$ is the loss of the cryostat and $P_{cl}$ represents the losses in the current leads. All of these losses are calculated in [W].

The AC losses of an HTS tape can be determined from the transport current and the transversal magnetic fields and are dominated by the hysteresis at the power network frequencies. The axial and radial components of the magnetic induction can be used as an input to determine the per unit length of the losses at the different part of the windings. The sum of the per unit AC losses of a BSSCO tape can be determined by the sum of the self, parallel, and perpendicular magnetic fields [24,50–52].

The self-field losses are given by the following formulae:

$$p_{sf} = I_c^2 \frac{f\mu_0}{\pi}[(1-\alpha)ln(1-\alpha) + \alpha - 0.5\alpha^2], \tag{15}$$

$$\alpha = \frac{I}{I_c}, \tag{16}$$

where the $I$ represents the nominal and $I_c$ the critical current in the superconducting tape, $f$ is the frequency and $p_{sf}$ denotes the self-field loss of the unit length of the superconducting

cable in W/m (Figure 4). The losses, due to the parallel field, can be described by the following formula [24,50]:

$$p_{||} = \frac{2fCA_c}{3\mu_0 B_p} B_{||}^3 \qquad\qquad B_{||} \leq B_{\perp} \qquad (17)$$

$$p_{||} = \frac{2fCA_c B_p}{3\mu_0}[3B_{||} - 2B_p] \qquad\qquad B_{||} > B_{\perp} \qquad (18)$$

where the $C$ coefficient represents a filling factor, to calculate the effective area from full cross-sectional area of the tape ($A_c$), and $B_p$ is the flux density of the full penetration field [27].

The unit length perpendicular field losses approximated by the following formula [24,50]:

$$P_{\perp} = K\frac{fw^2\pi}{\mu_0}B_c B_{\perp}[\frac{2}{\beta}log(ch(\beta)) - th(\beta)] \qquad (19)$$

where $K$ is a geometrical factor, and $B_c$ is a critical magnetic field of the tape, $\beta = \frac{\beta_{\perp}}{\beta_c}$.

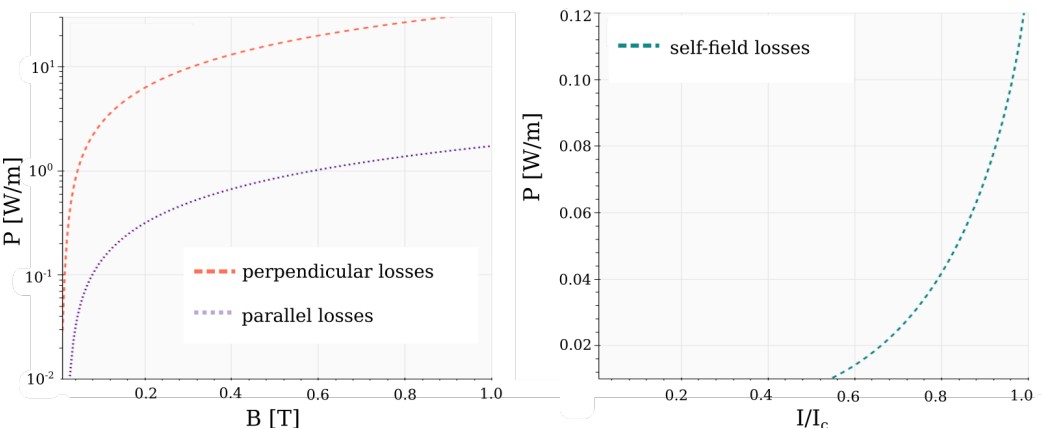

**Figure 4.** AC loss components of the superconductor due to the parallel and the perpendicular field losses and the self-field losses.

There is a major heat ingress created into the cryostat via the current leads. In case of a three-phase transformer, there are six current leads, where the heat leak for each metal lead is calculated by $q_{cl} = 42$ W/kA [53]. For instance, in [53] the measured 1 MVA superconducting transformer heat losses are estimated about 345 W, while its AC losses are about 360 W. For three phase transformers, the following formulas are used for calculating the transformer's thermal incomes [53]:

$$P_{cl} = n_{cl} \cdot q_c l \cdot (I_{plv} + I_{phv}), \qquad (20)$$

where $P_{cl}$ represents the thermal loss of the current leads in [W], $I_{plv}$ and $I_{phv}$ means the line and the phase currents in the low and the high voltage windings in [kA] and $n_{cl}$ means the number of the current leads in our case.

The losses of the cryostat can be calculated from its surface [30]:

$$P_{cr} = k_{th}/d_{th} \cdot A_{cr} \cdot 10^{-6} \cdot dT, \qquad (21)$$

where $k_{th} = 2.0 \cdot 10^{-3}$ W/(mK) represents the heat conductivity in the cryostat, $d_{th}$ is the thickness of the cryostat, $dT$ is the temperature difference between the inside temperature of the cryostat and the external temperature. Its value is $dT = 293 - 65 = 228$ K considering 20 °C outside temperature.

### 2.7. Calculation of the Short-Circuit Impedance and the Forces

During the optimization process, thousands of FEM simulations should be performed to evaluate all possible solutions. Therefore, the applied FEM model should also be simple and accurate. A two-dimensional, axisymmetric model of the transformer's working window was used to model the leakage flux distribution. This modelling technique has been used for decades in the transformer industry [54]. Despite its simplicity, its results meet the requirements for manufacturing accuracy. This model is generated from the independent model variables and solved by Agros Suite [41]. Due to the integrated advanced hp-adaptive FEM methods, this solver can resolve this simple FEM model within seconds with the required precision [48]. During the FEM analysis, the following Poisson equation is solved:

$$\Delta \vec{A} = \mu \vec{J}, \tag{22}$$

where $\vec{A}$ indicates magnetic vector potential, $\vec{J}$ is the vector of the current density in the coils and $\mu$ is the magnetic permeability. The magnetic core is defined by its relative permeability ($\mu_r$). In the simulations $\mu_r$ = 10,000 was applied, according to [48], while the boundary conditions of the problem are Dirichlet type ($\vec{A} = 0$). Figure 5 illustrates this calculation on the validated example, it shows the calculated results and a generated two-winding model.

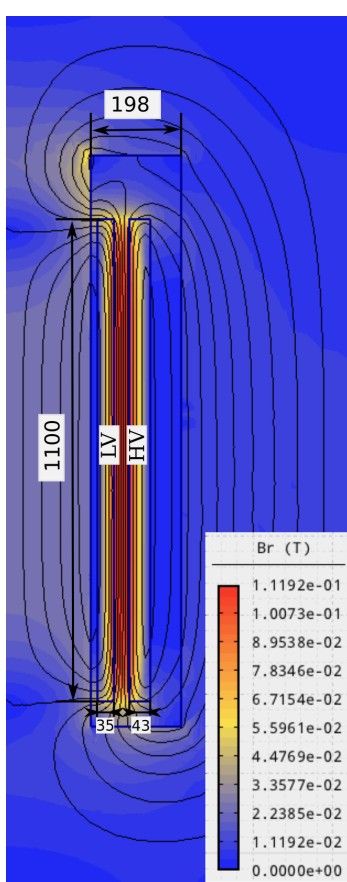

**Figure 5.** Magnetic flux distribution and the main dimensions of the transformer's working window.

Using the stored magnetic energy in the calculated area, the working window, the short-circuit reactance can be given in the following form [29,40]:

$$x_{\text{pu}} = \frac{\pi^2 f}{NI} \int_V \vec{J} \vec{A} \mathrm{d}V, \tag{23}$$

in this formula, $NI$ represents the sum of the amperturns in the windings. The $B_{ax}$ and $B_z$ values can be calculated for the different segments of the windings. In our calculations,

both the low-voltage and the high voltage windings, split into 20 different segments in the axial directions, were using its $B_{ax}$ and $B_{rad}$ values, the forces and the losses can be determined automatically in the model.

## 3. Results

### 3.1. Modelling and Optimization of a Conventional Transformer

The physical correctness and the accuracy of the applied transformer model and the implemented methodologies are demonstrated on a manufactured, three-phase, 10 MVA, 33/6.9 kV, star/star connected transformer. The transformer's datasheet and the detailed comparison with the measurements were originally published in [48]. The required and the independent parameters of a two-winding transformer model are presented in Table 3. All of the presented data can be accessed at TrafoCalc's repository [55].

**Table 3.** The required and the design parameters of the manufactured 3 phase, 10 MVA conventional power transformer.

| Requirements | Notation | Dimension | Value |
|---|---|---|---|
| Nominal Power | S | MVA | 10 |
| Frequency | f | Hz | 50 |
| Short-circuit impedance | SCI | % | 7.5 |
| Core filling factor | $ff_c$ | % | 89.2 |
| **HV winding** | | | |
| Connection type | - | - | star |
| Line Voltage | $U_{HV}$ | kV | 33 |
| Filling Factor | $ff_{HV}$ | % | 60 |
| **LV winding** | | | |
| Connection type | - | - | star |
| Line Voltage | $U_{LV}$ | kV | 6.9 |
| Filling Factor | $ff_{LV}$ | % | 70 |
| End insulation | EI | mm | 220 |
| Ratio of the LV/HV winding heights | $\alpha$ | - | 0.97 |
| Core gap distance | $g_{core}$ | mm | 20 |
| Phase distance | pd | mm | 50 |
| **Design Parameters** | | | |
| Core diameter | $D_c$ | mm | 420 |
| Flux density | $B_c$ | T | 1.7 |
| Current density in LV | $j_{lv}$ | $\frac{A}{mm^2}$ | 2.65 |
| Current density in HV | $j_{hv}$ | $\frac{A}{mm^2}$ | 2.57 |
| Height of the LV winding | $h_{lv}$ | mm | 1100 |
| Main insulation distance Costs | g | mm | 50 |
| No-load loss cost | $k_1$ | €/kW | 7100 |
| Load loss cost (I) | $k_{21}$ | €/kW | 1000 |
| Load loss cost (II) | $k_{22}$ | €/kW | 6000 |
| LV winding material | $c_1$ | €/kg | 10.0 |
| HV winding material | $c_2$ | €/kg | 9.5 |
| Core material cost | $c_0$ | €/kg | 3.5 |

The turn voltage is one of the firstly set parameters during a transformer design process, its value determines the ratio of the core and the windings, and many parameters, such as the number of the turns, depend on this value. Therefore, this is the first parameter calculated during the result analysis and the validation of the optimization model [3,44]. In the model, the turn voltage is calculated from the nominal power of the transformer, not directly with

the turns, which are the derived parameters of these values. The given value of the turn voltage (46.64 V) agrees with the measured value (46.87 V). A more precise calculation needs a detailed or a final design model from the proposed transformer and the windings. However, this detailed layout of the windings was unknown at the beginning of the preliminary design of the transformer and the goal of to find these most minute details in the later design process.

Another important parameter of every transformer is the short circuit impedance, whose value depends on the geometry of the modelled windings and the excitation. Its required value depends on the application, and even relatively small asymmetries in the winding system and imbalances in the excitation can complicate the calculation and decrease the efficiency of the analytical formulas. Many analytical and FEM-based methodologies have been developed to resolve this problem quickly and with the required accuracy [56]. In our case, the short circuit impedance is calculated by the integrated analytical formulae (SCI = 7.38%, [29]) and a FEM based calculation (SCI = 7.5%), both calculations are in very good agreement with the measured reference values (7.34% [48]).

The working window details and the magnetic flux distribution are depicted in Figure 5. The calculated height of the working window (1380 mm) agrees with the reference. The calculates mass of the transformers core is calculated as $M_c$ = 7751 kg, with this value the methodology approximates the core losses as $P_{nll}$ = 8.2 kW, with the built-in characteristic of the magnetic steel Figure 3.

The winding dimensions calculated from the independent parameters. The inner radius of the HV winding agrees well with manufactured transformers data, in which the inner diameter is 315 mm, while its outer diameter is 718 mm. The inner radius of the LV winding of the measured transformer model is 230 mm, while the outer radius is 265 mm. Therefore the model calculates these values well from the given parameters. The estimated AC loss in this winding is given by Kulkarni [48]: 489 W/phase, while the code calculates 1.44/3 = 480 W for the losses. The DC losses are approximated by the transformer coil's filling factor. The total loss of the LV windings in the reference is 21.8 kW [48], while the algorithm calculates: $P_{LV} = P_{DC} + P_{eddy}$ = 19.9 kW + 1.45 kW = 21.35 kW.

### 3.2. Optimization and TOC Calculations for Two Distinct Scenarios

The proposed transformer models can insert a transformer optimization chain directly using the Ārtap-framework [40,42]. The previously created 10 MVA transformer model was optimized for two different economic scenarios. These two scenarios were selected from [7], where the no-load loss cost of the transformer was given as $k_1$ = 7000 €/kW, while the cost of the load losses differs significantly due to the differences in the utilization factor. In the first case, the transformer joints a PV plant, where the cost of the load losses is calculated by $k_2$ = 1000 EUR/kW, while in the second case, the transformer joints a biomass power station, which continuously produces electricity. In this case the capitalization cost of the load losses are calculated by $k_2$ = 6000 EUR/kW.

The widely used NSGA-II algorithm is performed for the optimization, which is one of the most popularly-used genetic algorithm-based, multi-objective optimization techniques due to its good convergence stability [57,58]. We used the implementation of this algorithm from the Ārtap-framework [58,59]. The optimization search calculated 30 generations of individuals and distinct transformer designs in 30 populations. The current density of the transformer windings varied from 2.0 $\frac{A}{mm^2}$ to 3.2 $\frac{A}{mm^2}$, considering the heat flux limitations in the transformer windings. The main insulation distance between the LV and HV windings varied from 20 mm to 60 mm, and the LV winding's height varied between 800 mm and 1400 mm. Considering the buckling stress, the diameter of the core should be limited; therefore, the core diameter of the transformer's core should be smaller than 360 mm, and the flux density should be set between 1.5 T and 1.7 T because there is no noise or other restrictions which should limit this value. The results of the two optimizations are summarized in Table 4.

**Table 4.** The optimized parameters and the capitalization costs of the manufactured and the two optimized designs.

| Parameter | Dimension | Manufactured | Scenario I | Scenario II |
| --- | --- | --- | --- | --- |
| Core diameter | mm | 420 | 405 | 485 |
| Flux density | T | 1.7 | 1.56 | 1.57 |
| Current density in LV | $j_{lv}$ | 2.65 | 2.86 | 2.1 |
| Current density in HV | $j_{hv}$ | 2.57 | 2.33 | 2 |
| h_in | mm | 1100 | 1025 | 952 |
| m_gap | mm | 50 | 20 | 55 |
| Capitalization cost | | | | |
| Scenario I | € | 161,503 | 147,893 | 167,987 |
| Scenario II | € | 407,403 | 419,588 | 347,037 |
| Core loss | kW | 8.2 | 5.5 | 8.9 |
| Load loss | kW | 49.18 | 54.34 | 35.81 |

Some extreme example of the capitalization costs was selected to demonstrate how important to consider the application and the designed transformer together from the beginning of the transformer design. Not enough to consider the efficiency of the transformer only to produce an economical design. The results show that two very different design conceptions will be the winning in the two selected scenarios. None of them is better than the reference design in both scenarios.

In the case of the first scenario, due to the low load loss price, the transformer has a 17% smaller core and about 50% smaller current densities than in the second scenario, where both of the capitalization costs are high due to the smaller core and windings, which can increase the core mass via the window width of the transformer. The second scenario's optimal design is more energy efficient than the other. However, it can be seen from the results that none of the transformers' designs is better than the other. If we are comparing the prices with the same capitalization levels, the low-loss transformer has a 13.5% higher lifetime cost in the first scenario than the high loss design, while the high loss transformer design has a 20.5% higher lifetime cost in the second scenario.

*3.3. Analysis of a Superconducting Transformer*

The applied formulas and modifications compared to the superconducting model are tested during the analysis of a 1250 kVA, 10.5 kV/0.4 kV, YNy0 connected three-phase superconducting transformer, whose detailed analysis and main parameters are presented in [19]. The required and the independent parameters of a two-winding transformer model are presented in Table 5.

The turn voltage of the transformer calculated back from the given data is given 23.12 V-s. Therefore it gives back the reference values. The calculated short circuit impedance (FEM calculated SCI = 5.44%, analytical formula calculated SCI = 5.29%) is smaller than the given data in the paper. However, this difference can be acceptable. The main reason is that the authors used a special winding arrangement, where the top and the bottom pancakes of the HV winding are much wider than the others. This can cause a difference compared to the simple, automatically generated two-winding transformer model.

The calculated values of the LV (inner radius = 205.1 mm, thickness = 13.5 mm, winding height = 342.5 mm) and HV winding (inner radius = 253.1 mm, thickness = 8.2 mm, winding height = 356.2 mm) agree with the published data of the reference paper. The AC losses of the superconducting windings are calculated by using the FEM calculated axial ($B_{ax}$) and radial ($B_{rad}$) flux density components. The LV and the HV windings are divided into 21 segments, where these values are calculated separately. This is the difference between the previously applied loss calculation models, where the analytically calculated maximum value of the axial and radial fluxes was used for the loss calculations. To compare

the difference between the two methodologies, both the two calculations were made during the analysis. The built-in methodology, which calculates the losses with the sum of the 21 segments, results in $P_ac$ (avg) = 248 W, while the other methodology, which uses only the maximum value of the windings, results in $P_ac$ (max) = 1540 W. The reason for this significant difference can be seen in Figure 6, which shows that the maximum values of the radial and the axial flux densities are much larger than the average of these values along the windings.

**Table 5.** The required and the design parameters of the three phase 1250 kVA transformer.

| Requirements | Notation | Dimension | Value |
|---|---|---|---|
| Nominal Power | S | MVA | 1.250 |
| Frequency | f | Hz | 50 |
| Short-circuit impedance | SCI | % | 5.8 |
| Core filling factor | $ff_c$ | % | 92.0 |
| **HV winding** | | | |
| Connection type | - | - | star |
| Line Voltage | $U_{HV}$ | kV | 0.4 |
| Filling Factor | $ff_{HV}$ | % | 8.7 |
| **LV winding** | | | |
| Connection type | - | - | star |
| Line Voltage | $U_{LV}$ | kV | 10.5 |
| Filling Factor | $ff_{LV}$ | % | 11.6 |
| End insulation | EI | mm | 481 |
| Ratio of the LV/HV winding heights | $\alpha$ | - | 1.04 |
| Core gap distance | $g_{core}$ | mm | 50 |
| Phase distance | pd | mm | 30 |
| **Design Parameters** | | | |
| Core diameter | $D_c$ | mm | 310 |
| Magnetic Flux density | $B_c$ | T | 1.5 |
| Current-density in LV winding | $j_{lv}$ | $\frac{A}{mm^2}$ | 44.92 |
| Current-density in HV winding | $j_{hv}$ | $\frac{A}{mm^2}$ | 53.45 |
| Height of the LV winding | $h_{lv}$ | mm | 342.5 |
| Main insulation distance | $g$ | mm | 34.5 |

In the case of superconducting transformers, the major component of the load losses is generated by the current leads ($P_{cl}$), and a significant amount of cooling power should be added to consider the losses of the cryostat ($P_{cr}$) [15]. During the analysis, the heat leak of each metal heat lead is calculated as 42 W/kA [15]. By their nominal current, six current leads are considered to estimate the heat income. The calculated thermal losses are given 505.71 W for the three phases, and this relatively large number agrees with other sources. For instance, in [53], the measured 1 MVA transformer heat losses are estimated at about 345 W, while the AC losses are about 360 W. This quantity should be higher in this case due to the higher currents.

The third part of the load losses comes from the cryostat losses, which are calculated from the estimated surface of the cryostat and the previously shown thermal conductivity (Equation (21)). The code resulted $P_{cr}$ = 42.54 W for the three phases. Therefore, the load losses in the given superconducting results given (Equation (14)) $P_{ll}$ = 252 + 42.54 + 505.7 = 800 W, which is in good agreement with the results of the reference transformer [19].

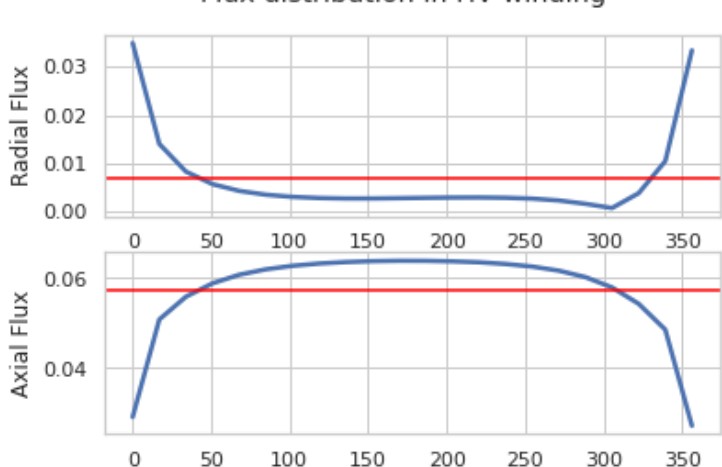

**Figure 6.** The figure shows the results of finite element method-based calculation of the axial and radial components of the magnetic flux density along the high voltage winding of the modeled transformer. The red line shows the average value of the radial and the axial magnetic flux along the winding.

## 4. Conclusions

This paper introduced a novel two-winding transformer model, which can be easily used to calculate the main dimensions of conventional and superconducting transformer designs. Both superconducting and conventional transformer models can be defined as key-value pairs in a uniform JSON format. The TrafoCalc library automatically calculates the key parameters of the transformer thus defined. This JSON-based workflow can be easily integrated–directly or as an API—into an evolutionary algorithm-based optimization loop.

The proposed two-winding model contains an integrated, hp-adaptive solver-based, two-dimensional FEM model to determine the magnetic flux distribution in the working window of the transformer and estimate the short circuit impedance and the axial and radial flux distribution along the windings. The algorithm uses the resulting magnetic flux distribution to estimate the losses in the transformer windings both in the conventional and superconducting case. Therefore the superconducting loss calculation differs from many previously published optimization algorithms, which only calculated the maximum value of the radial and the axial flux density in the windings, thus over-estimating the AC losses of the superconducting winding. As shown in the validation example, the proposed method resolves this problem and makes a basis for more realistic preliminary design calculations in the superconducting case. The proposed conventional transformer model was validated by the measured data of 10 MVA transformer. Then a simple optimization was performed to show the capabilities of the proposed TrafoCalc library (https://github.com/tamasorosz/TrafoCalc/, accessed on 28 May 2022) and the effect of the capitalization factors on the optimal transformer design. This small optimization task illustrated that not always better efficiency is the key to creating an economical transformer design. In the case of a PV scenario, the estimated lifetime cost of the high-loss transformer is significantly smaller.

The superconducting transformer calculations were validated on a three-phase, 1.2 MVA transformer, while its cost optimization and realistic comparison with a conventional transformer are planned to be compared in a future paper. The proposed calculations and the source of the python library are available at the project's Github repository (https://github.com/tamasorosz/TrafoCalc/, accessed on 28 May 2022).

**Funding:** This research received no external funding.

**Institutional Review Board Statement:** Not applicable.

**Informed Consent Statement:** Not applicable.

**Data Availability Statement:** Not applicable.

**Conflicts of Interest:** The author declares no conflict of interest.

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
