# Peer review of "FEM-Based Power Transformer Model for Superconducting and Conventional Power Transformer Optimization"

_energies, doi:10.3390/en15176177_

Round 1

Reviewer 1 Report

There are 3D computer models of conventional oil transformers, which are not discussed in the article and are missing in the references.

Power losses when loading a transformer also depend on the temperature. On the other hand, the temperature of the transformer depends on the power losses in it and the method of cooling. Therefore, it is necessary to solve the coupled electromagnetic-thermal problem. It is not clear how the created model based on FEM takes into account the thermal processes in the transformer – heat conduction, radiation and convection. How is the coefficient of heat transfer from the surface of the transformer to the environment determined? It is also not clear which software product was used.

Besides, there are some other inaccuracies in the article: in formula 4 Mcorner is written, while later in the explanation Mcore is used instead; there are no commas after the formulae (1), (2), (3), (8), (10), (16), (17), (18), and (22); in rows 143 and 183 the word “where” should be lowercase; the explanations below the formulae should also be lowercase and not indented; unclear writing exists in row 174 - [35? ]; the variable Acu in formula (10) is not explained etc. 

Author Response

I would like to thank the Reviewer for his work and his questions to help me improve the paper.

There are 3D computer models of conventional oil transformers, which are not discussed in the article and are missing in the references.

Thank you for this comment. The role of the paper is to present a preliminary design model, which thousands of calculations should consider in a very short time. In this designing phase, the engineer's goal is to make a quotation for a customer, where he has time to calculate a simplified transformer model within 1-3 days. This is the reason why I missed these references from this paper. I have added some more explanations to the beginning of section 2.7 to clarify this issue.

Power losses when loading a transformer also depend on the temperature. On the other hand, the temperature of the transformer depends on the power losses in it and the method of cooling. Therefore, it is necessary to solve the coupled electromagnetic-thermal problem. It is not clear how the created model based on FEM takes into account the thermal processes in the transformer – heat conduction, radiation and convection. How is the coefficient of heat transfer from the surface of the transformer to the environment determined? It is also not clear which software product was used.

You are right. The IEC – 60076 standard simplifies this question because it defines the outside temperature as 40°C, and the load losses should be considered at 75° at the nominal stage of the power transformers. Therefore, the first answer is simple; the resistivity value is selected according to the temperatures mentioned above. The second part of your question is interesting because the calculation of the radiation, convection and conduction is not so simple task. In this model its simply considered by κ in this phase, which is an empirical factor but gives a good approximation for the extra losses of the transformer tank. This factor will be more complicated if the transformer contains magnetic shielding, but it works in this preliminary design case.

Besides, there are some other inaccuracies in the article: in formula 4 Mcorner is written, while later in the explanation Mcore is used instead;

Corrected.

there are no commas after the formulae (1), (2), (3), (8), (10), (16), (17), (18), and (22);

Corrected.

in rows 143 and 183 the word “where” should be lowercase; the explanations below the formulae should also be lowercase and not indented; unclear writing exists in row 174 - [35? ]; the variable

Corrected.

Acu in formula (10) is not explained etc.

Corrected.

Reviewer 2 Report

The article presents the FEM computation and analytical optimization of design of conventional and superconductiong power transformers. The subject of the article is interesting, however I find a few drawbacks:

1. The FEM model, as a contrary to the title of the article suggests, is almost no explained. I have no idea how the FEM model is prepared and simplified. Especially it would be useful for the superconducting-winding transformer. 

2. The superconducting-winding transformer description is very scarse and the results are very general. E.g. how the model of cryostat was prepared?

3. The introduction suggests that the article is going to show the economical comparison of both types of transformers (traditional vs. superconducting), however I do not find it in the article. 

4. The FEM models were validated with the "measured" references, according to the Author. However, the goal (and novelty) of this manuscript is the technical-economical set of equations presented in the Chapter 2. How was this model (equations) validated? In my opinion, the Author should prepare validation of his new algorithm and compare it with currently existing. 

Some detailed comments:

Line 331, 125 and whole article: The number of figure should be in the proper order i.e. Fig. 1, 2, 3... . Here we start counting with fig. 4. 

Some figures are not explained or even not mentioned in the text (e.g. Fig. 1). They must be mentioned and explained in the text otherwise they are useless and should be removed. 

Table 1: There should be reference to each of the construcion (the constructions are explained in scientific articles). 

Even in the title of the article the "FEM" model is mentioned however, one of the most significant paragraph 2.1 contains very little informaion about FEM model of the transformer. For example, it is very difficult to compute full model of such a big transformer due to the number of mesh elements. Was there used any symmetry? Was the winding simplified anyhow? I do not find Litz losses anywhere, how were they computed? 

Chapter 2.6: The chosen REBCO tape should be explained here in deails. Moreover, from the text I understand, that the Author assumed that superconducting winding is made of very simple pattern. However, today there are used other types such as e.g. Roebel Cables, which significantly reduce AC loss. What is more, here the Author assumed only parallel and perpendicular fields while the REBCO can be immersed in the external field of different direction. It is especially important while artificial pinning centers (APC) are created in the tape (DOI:10.1109TASC.2018.2854673). Does your tape contain APC? 

Table 3: I do not fully understand k1, k21 and k22. Are these prices per year or per life-time? BTW, in my opinion their unit is incorrect because it should contain time or the price should be given per amount of energy instead of power. 

Lines 259-260: which measured value? Up to this line, the reference 'measured' transformer was not even mentioned. 

Lines 260-262: "The more precise agreement needs more precise knowledge of the detailed winding layout and its copper filling factor, which was unknown at the beginning of the preliminary design." - I do not understand. So what was the base of the measured transformer? Maybe it is more convinient to chose better described model?

Fig. 5. I do not understand this figure and the caption. What does exacly mean "some main parameters" ? What is the purpose of showing of magnetic flux disribution in the windings? Moreover, where is the core in this simulation?

Lines 269-277:  Why do you compute AC loss according to [45], while you have FEM model? 

Line 285 and 287: Those costs 1000 €/kW and k2 = 6000 €/kW are incomprehensible. The price should be per amount of energy not a power. Otherwise, there should be explenaion of the time (per nominal power per year?). 

The superconducing transformer model is too generally described. Line 345: I do not understand this equation in line. 

I do not see any cost information associated with superconducting winding transformer. I do no see comparison of the superconducting and traditional transformer. 

Author Response

I would like to thank the Reviewer for their work and valuable comments, which helped me improve the paper.

1. The FEM model, as a contrary to the title of the article suggests, is almost no explained. I have no idea how the FEM model is prepared and simplified. Especially it would be useful for the superconducting-winding transformer.

The applied FEM model is a 2d axisymmetric model, which uses the hp-adaptive FEM solver, Agros2D to calculate its parameters. The details of this model are described in section 2.7. I have revised the text to be more descriptive for the reader. Moreover, the detailed calculation is presented in the connecting Jupyter notebook, which can be available at the project's Github repository. All of the presented calculations and the connecting code can be downloaded from the Github repository of the project.

2. The superconducting-winding transformer description is very scarse and the results are very general. E.g. how the model of cryostat was prepared?

The basic idea of the model is to use the copper filling factor-based conventional transformer model for superconducting transformers. These conventional transformers also have many winding systems, which have many advantages for short circuit stability, better cooling properties, and faster manufacturing. However, during the very first design step, where the engineer has a very short time (1-3 days) to select the optimal key parameters, there is no time for creating a final design. The goal of this preliminary design stage is to create a coarse model, which is detailed enough to select the most economical solution. The filling factor-based models can be used with good accuracy. This is the goal of this paper to generalize this model for superconducting cases, as well.

The cryostat model in this very first approximation is only modelled as its approximative main dimensions (thickness) and a thermal coefficient. The detailed model of the cryostat can be modelled during the final design phase, not this conceptional comparison.

3. The introduction suggests that the article is going to show the economical comparison of both types of transformers (traditional vs. superconducting), however I do not find it in the article.

The goal of the paper is only to show the proposed Python library, which contains a two-winding transformer model which can be applied to calculate the main design parameters of a conventional and a superconducting transformer. This is the novelty of the paper because the previous studies only optimized a superconducting transformer and compared this design with a selected conventional transformer. This comparison is not correct because different designs can be optimal in different economic situations. The proposed model can be used to calculate the preliminary/quotation design, which is a rough model. It can be used as an equal basis of the comparison of the transformer models.

4. The FEM models were validated with the "measured" references, according to the Author. However, the goal (and novelty) of this manuscript is the technical-economical set of equations presented in the Chapter 2. How was this model (equations) validated? In my opinion, the Author should prepare validation of his new algorithm and compare it with currently existing.

The objective function of the techno-economic comparison is on the standard practice. Please, check IEC 60076-20: 2017, Power Transformers – Part 20, or the other referenced literature. This formula extended to the superconducting transformers. The conventional transformer optimization code is based on previously published works, the role of this paper to extend this code to can handle superconducting transformers, as well. The proposed equations can be checked in the projects GitHub repository, where every calculation tested, most of them validated by referenced examples. 

The detailed calculation and the measured values of the 10 MVA conventional transformer are coming from section 3.1, where in reference [48] was given. All of the calculations can be checked in the proposed library, all of the proposed calculations unit tested, you can check it here: https://github.com/tamasorosz/TrafoCalc/. You can check the superconducting transformer model and its references in the library.

Some detailed comments:

Line 331, 125 and whole article: The number of figure should be in the proper order i.e. Fig. 1, 2, 3... . Here we start counting with fig. 4.

Corrected.

Some figures are not explained or even not mentioned in the text (e.g. Fig. 1). They must be mentioned and explained in the text otherwise they are useless and should be removed.

Corrected.

Table 1: There should be reference to each of the construcion (the constructions are explained in scientific articles).

The used references cited in the paper, the role of this table is to show some existing projekt and the importance of the topic.

Even in the title of the article the "FEM" model is mentioned however, one of the most significant paragraph 2.1 contains very little informaion about FEM model of the transformer. For example, it is very difficult to compute full model of such a big transformer due to the number of mesh elements. Was there used any symmetry? Was the winding simplified anyhow? I do not find Litz losses anywhere, how were they computed?

The applied FEM model is a 2d axisymmetric model, which uses the hp-adaptive FEM solver, Agros2D, to calculate its parameters; the details of this model are described in section 2.7. I have revised the text to be more descriptive for the reader. Moreover, the detailed calculation is preseneted in the connecting Jupyter notebook, which can be availabe at the Github repository of the project. All of the presented calculations and the connecting code can be downloaded from the Github repository of the project.

Chapter 2.6: The chosen REBCO tape should be explained here in deails. Moreover, from the text I understand, that the Author assumed that superconducting winding is made of very simple pattern. However, today there are used other types such as e.g. Roebel Cables, which significantly reduce AC loss. What is more, here the Author assumed only parallel and perpendicular fields while the REBCO can be immersed in the external field of different direction. It is especially important while artificial pinning centers (APC) are created in the tape (DOI:10.1109TASC.2018.2854673). Does your tape contain APC?

The proposed reference (DOI:10.1109TASC.2018.2854673) cited in the paper.

The basic idea of the model is to use the copper filling factor based conventional transformer model for superconducting transformers. These conventional transformers also has many winding systems, which has many advantages for short circuit withstand stability or better cooling properties, or faster manufacturing. However, during the very first design step, where the engineer has 1-3 days to select the optimal key parameters, there is no time for these complex calculations. The filling factor based models can be used with good accuracy, this is the goal of this paper to use this model for superocnducting case, as well.

Table 3: I do not fully understand k1, k21 and k22. Are these prices per year or per life-time? BTW, in my opinion their unit is incorrect because it should contain time or the price should be given per amount of energy instead of power.

The calculation is correct and follows the standard : IEC 60076-20: 2017, Power Transformers – Part 20. The detailed calculation of the losses explained in the given reference. The unit is correct, because all of the losses calculated for the transformers planned lifetime, which is usually 20-30 years. This technique is widely used in the industry for evaluating quotations.

Lines 259-260: which measured value? Up to this line, the reference 'measured' transformer was not even mentioned.

The previous lines at the beginning of the chapter (250-253) contains the description and the referecnes of the measured transformer:

„The accuracy and the physical correctness of the applied transformer model and the implemented methodologies is demonstrated on a manufactured, three-phase, 10 MVA, 33/6.9 kV, star/star connected transformer. The transformer's datasheet and the detailed comparison with the measurements were originally published in [46].”

The calculation and the main parameters of the transformers can be downloaded from the projekts’ Github repository: https://github.com/tamasorosz/TrafoCalc/

Lines 260-262: "The more precise agreement needs more precise knowledge of the detailed winding layout and its copper filling factor, which was unknown at the beginning of the preliminary design." - I do not understand. So what was the base of the measured transformer? Maybe it is more convinient to chose better described model?

The basis of the comparison and the measured transformer was referenced in the previous sentences.

The more precise calculation needs a detailed or a final design model from the proposed transformer and the windings, not only a copper filling factor-based transformer model. However, this detailed layout of the windings is unknown at the beginning of the examined preliminary design of the transformer.

Fig. 5. I do not understand this figure and the caption. What does exacly mean "some main parameters" ? What is the purpose of showing of magnetic flux disribution in the windings? Moreover, where is the core in this simulation?

Thank for the comment. The caption of the figure corrected: Magnetic flux distribution and the main dimensions of the transformer's working window. 

The role of the figure is to illustrate the used 2d axisymmetric working window model of the transformer. This technique is well known and widely used in the industry, it is described in detail in the previously asked reference [45]. 

Lines 269-277: Why do you compute AC loss according to [45], while you have FEM model?

Reference [45] describes the applied FEM-based calculation of AC losses in transformer windings, this is a standard practice and this simple methodology can be used to estimate accurately the losses, while only the filling factor of the windings are known not the type and the layout of the windings. 

Line 285 and 287: Those costs 1000 €/kW and k2 = 6000 €/kW are incomprehensible. The price should be per amount of energy not a power. Otherwise, there should be explenaion of the time (per nominal power per year?).

No, this is the standard practice in the transformer industry. Please check IEC 60076-20: 2017, Power Transformers – Part 20, or the referenced literature. The costs and the applied units are correct. The goal of this selection is to show that two different designs can be optimal in very different economic case.

The superconducing transformer model is too generally described. Line 345: I do not understand this equation in line.

I do not see any cost information associated with superconducting winding transformer. I do no see comparison of the superconducting and traditional transformer.

The goal of the paper is only to show the proposed Python library, which contains a two-winding transformer model, which can be applied to calculate the main design parameters of a conventional and a superconducting transformer. This is the novelty of the paper, because the previous studies only optimized the conventional transformer, or only a superconducting transformer. The proposed model can be used to calculate the preliminary/quotation design, which is a rough model, it is generally used as a basis of the comparison of the transformer models.

Round 2

Reviewer 2 Report

Dear Author, thank you for your responses. 

Line 278: The hyperlink should be provided as a reference number instead of "in line". 

I still do not see the Litz losses estimation in the manuscript. They are very important for high-frequency transformers or high-current (power) transformers. They should be estimated. 

Author Response

Dear Reviewer,

 Thanks for your questions and work to help improve the quality of the paper.

Line 278: The hyperlink should be provided as a reference number instead of "inline". 

Corrected.

I still do not see the Litz losses estimation in the manuscript. They are very important for high-frequency transformers or high-current (power) transformers, and they should be estimated. 

Thanks for this comment, section 2.5 changed to  answer your question.
The code can use two methodologies to calculate the Litz-losses in the windings (in the paper, these losses are considered and referred to as a part of "eddy losses"). In the applied methodologies discussed in section 2.5, the first approximation uses only a number, which is estimated from the shape of the winding to approximate the losses, while the second approach uses the resulting magnetic flux distribution to estimate the winding losses. This approach assumes that the windings do not modify the magnetic field distribution around them. This is not true, as you mentioned, in the case of a furnace transformer or another transformer with very high currents in the windings. However, this methodology gives good results in comparison with the measurements.